Brief Investigation

# The illusion of polygenicity in pool-seq genetic mapping studies: insufficient power can mask simple genetic architectures

Anthony D. Long,[1,]* Katherine M. Hanson [ID],[2] Stuart J Macdonald [ID][2]

[1]Ecology and Evolutionary Biology, University of California, 321 Steinhaus Hall, Irvine, CA 92697, United States
[2]Department of Molecular Biosciences, University of Kansas, 1200 Sunnyside Avenue, Lawrence, KS 66045, United States

*Corresponding author: Ecology and Evolutionary Biology, University of California, 321 Steinhaus Hall, Irvine, CA 92697, United States. Email: tdlong@uci.edu

Pool-seq (pooled sequencing) combines DNA from multiple individuals prior to sequencing, enabling population-level allele frequency estimation without individual genotyping. When employed in a case-control genome-wide association study (GWAS) framework, pool-seq faces a fundamental power limitation: Errors on allele frequency estimates are inversely proportional to sequence coverage and are large at modest coverage levels. Although this power limitation is appreciated, modestly sized pool-seq GWAS lacking unambiguous hits are often interpreted as showing a polygenic genetic architecture. We illustrate that this inference is unwarranted using empirical data from a *Drosophila* zinc resistance mapping study. Despite achieving >700× sequencing coverage in case and control pools, a directly ascertained SNP-based GWAS failed to reveal clear evidence for major-effect loci. A unique feature of the dataset is that an advanced intercross MPP, with known founders, was employed as the GWAS population. We leverage this unique population structure, in a manner that would not be possible in an outbred panel, to carry out 2 additional GWASs using imputed haplotype- or SNP-frequency estimates, which in contrast uncover localized regions of major effect. The key difference between approaches lies in statistical power: Directly ascertained SNP counts have errors inversely proportional to sequencing coverage, whereas known-founder imputation–based approaches can be considerably more accurate. In outbred populations where imputation cannot be used to obtain more accurate allele frequency estimates, substantially higher coverage than currently envisioned may be required to reliably detect modest allele frequency shifts. This work highlights that insufficiently powered GWAS can mask simple genetic architectures and create the illusion of polygenicity through statistical noise alone.

Keywords: pool-seq; X-QTL; *Drosophila*; Zinc; polygenic; power

## Introduction

Pool-seq (pooled sequencing) is a cost-effective genomic approach where DNA from multiple individuals is combined prior to short-read sequencing, allowing researchers to estimate the allele frequency in a population without individually genotyping each individual (Futschik and Schlötterer 2010; Schlötterer et al. 2014). Pool-seq has been widely employed across diverse research contexts including population genomics (Kofler et al. 2011; Czech et al. 2024), experimental evolution studies (Burke et al. 2010, 2014; Turner et al. 2011; Orozco-terWengel et al. 2012), agricultural breeding programs (Michelmore et al. 1991), and genetic mapping in model organisms (Ehrenreich et al. 2010; Macdonald et al. 2022). A fundamental principle of pool-seq is that, for autosomal loci, when the number of chromosomes in the pool, 2N, substantially exceeds the depth of sequence coverage, $C$ (i.e. $2N >> C$), directly ascertained SNP-frequency estimates provide unbiased estimates of allele frequencies in the population from which the sample is drawn (Futschik and Schlötterer 2010). Furthermore, the precision of these frequency estimates follows binomial sampling theory, with standard errors proportional to $\sqrt{(pq/C)}$, where $p$ and $q$ are the reference and alternative allele frequencies.

Pool-seq has been increasingly applied to case-control genome-wide association studies (GWASs), where researchers compare allele frequencies between phenotypically distinct pools of individuals to identify genetic loci contributing to complex trait variation (Huang et al. 2012; Bastide et al. 2013; Morozova et al. 2015; Fochler et al. 2017; Zhou et al. 2017; Macdonald et al. 2022; Macdonald and Long 2022). This approach is analogous to the human case-control GWAS design (Risch and Merikangas 1996; Wellcome Trust Case Control Consortium 2007), in which a phenotypically extreme group of $N_1$ affected case individuals is compared to $N_2$ control individuals, except that rather than genotyping individuals via SNP arrays, pools of DNA from case or control individuals are subjected to short-read sequencing to coverages $C_1$ and $C_2$. A standard GWAS tests for differences in allele frequencies between the $2N_1$ case and $2N_2$ control autosomal alleles, whereas under the pool-seq design, the tests are for differences between the $C_1$ case and $C_2$ control reads per locus. Several studies have successfully implemented this approach across model organisms demonstrating its feasibility for trait dissection (Craig et al. 2009; Bastide et al. 2013; Lirakis et al. 2022; Davies and Myles 2023). However, much like a standard GWAS, pool-seq GWAS faces the fundamental trade-off between controlling false-positive rates through stringent significance thresholds and

maintaining sufficient statistical power to detect true associations. Human GWAS studies employ −log10(P-value) significance thresholds exceeding 6 to 9 paired with greater than 3,000 fully genotyped case and control individuals to reliably detect subtle effect risk alleles while controlling for multiple tests.

The power limitations of case-control comparisons using pool-seq are well established (Baldwin-Brown et al. 2014; Kofler and Schlötterer 2014): To reliably detect allele frequency differences <10%, pools of thousands of individuals sequenced to >1000× average coverage are likely required. Despite this constraint, Manhattan plots of −log10(P-value) against genomic location from experiments with much lower sequence coverage that show little evidence of "hits" are often interpreted as representing a highly polygenic architecture (Huang et al. 2012; Orozco-terWengel et al. 2012; Morozova et al. 2015; Fochler et al. 2017; Zhou et al. 2017; Barghi et al. 2019; Lirakis et al. 2022). A polygenic architecture may indeed underlie the trait in these cases, but Manhattan plots from modestly powered studies are a poor proxy for genetic architecture. To quantify the nature and extent of this problem, we conduct simulations and reanalyze a recent *Drosophila* X-QTL study as though it were a directly ascertained case-control experiment. We show that even at sequence coverages exceeding those commonly employed, it is difficult to detect genes of large effect. More caution should be exercised in interpreting Manhattan plots from such studies.

## Methods

The zinc chloride mapping dataset is described in detail in Hanson et al. (Hanson et al. 2025). Briefly, the study utilized a population derived by intercrossing 663 *Drosophila* Synthetic Population Resource (DSPR) "A" founder population recombinant inbred lines. After 31–36 generations of synthetic population maintenance, 12 replicate zinc selection assays were performed. To perform the assays, 0- to 24-h-old embryos were collected from the base population and transferred to control (water) or treatment (25-mM zinc chloride) bottles. Emerged adult females were collected, and pooled DNA samples were prepared for both control and zinc-selected groups, DNA libraries were prepared using the Illumina DNA Prep protocol and sequenced with PE150 reads on an Illumina NovaSeq 6000. Reads were mapped to the *Drosophila melanogaster* Release 6 reference genome, and REF/ALT counts were obtained for every SNP. Founder haplotype frequencies were estimated from observed SNP frequencies and known-founder haplotypes in overlapping 1.5-cM windows using R/limSolve (Soetaert et al. 2009). limSolve uses a constrained linear optimization procedure for each window with the constraints defined such that all founders have frequencies between zero and one and the frequencies further sum to one. This approach for estimating founder haplotype frequencies is validated in Linder et al. (2020) and Macdonald et al. (2022).

Here, we employ the raw SNP tables and haplotype frequency estimates from Hanson et al. (2025). To enable direct comparison across analytical approaches, all statistical tests employ Cochran–Mantel–Haenszel (CMH) tests, which are appropriate for the 12-replicate control vs zinc-selected comparisons in this experiment. For the directly ascertained SNP scan, CMH tests use the observed REF vs ALT read counts at each SNP. For the imputation-based approaches (haplotype-level and SNP-level scans), we first impute haplotype frequencies in sliding windows using the constrained linear optimization approach described in Hanson et al. (2025) summarized above. SNP frequencies were then additionally imputed by multiplying the closest set of

estimated haplotype frequency estimates by the founder states (generally 0 or 1) for each haplotype. In both cases we convert frequencies to counts by multiplying by 2N, where N is the number of diploid individuals contributing to each pooled DNA sample, treating imputed frequencies as if they were obtained from directly genotyping N individuals. However, because haplotype frequency imputation from pool-seq is subject to uncertainty, the effective sample size is reduced relative to individual genotyping by some efficiency factor $k < 1$. Since chi-square test statistics scale linearly with the sample size, this overestimation inflates test statistics and consequently −log$_{10}$(P-values), by a constant genome-wide factor in the imputation-based scans (Fig. 3b and 3c). Based on comparisons to high-coverage validation data (Linder et al. 2020), we estimate this inflation factor to be approximately 30%. A key point is that this inflation factor is uniform across the genome and therefore does not affect the relative ranking of associations or the identification of peaks, though absolute P-value thresholds should be interpreted accordingly.

MSPrime (Kelleher and Lohse 2020; Baumdicker et al. 2022) was used to simulate 500 haplotypes from a neutrally evolving 1Mb region under *D. melanogaster*-like wild population parameters ($N_e$ = 1e6, $u$ = 5e−9, $r$ = 2e−8). These haplotypes were sampled with replacement to create a 10,000 haplotype control base population. A similar 10,000 haplotype case population was created by sampling haplotypes with replacement from the control population conditional on a single focal SNP having an allele frequency difference of 4% or 8% from the controls. Frequencies were then estimated at all common SNPs for both case and control populations, and then pool-seq allele counts obtained via random negative binomial draws (size = mu = expected coverage) conditional on the known frequencies and expected sequencing coverages of 400×, 1,000×, or 5,000×. The negative binomial was employed so that realized coverage varied among SNP positions according to an overdispersed Poisson distribution with a variance twice the mean (equal to expected coverage). Chi-square tests were carried out at each SNP and −log10(P-values) obtained.

## Results

We initially used simulations to quantify the limitation of pool-seq case-control studies to detect genes of large effect. We simulated experiments with pools of 10,000 individuals, varying levels of sequencing coverage (400×, 1,000×, and 5,000×), and a single SNP with a true allele frequency difference of 4% or 8% between pools (Fig. 1). Four-hundred-fold coverage could be routinely achieved in pool-seq experiments, while 5,000× although challenging is comparable in scale to a human GWAS study consisting of 2,500 fully genotyped samples. Our simulations suggest that allele frequency differences of 8% are only reliably detected at 5,000× coverage. Four to eight percent allele frequency differences would be considered large by the standards of human disease GWAS studies, yet the sample sizes and coverages required to detect these loci are never achieved in published pool-seq studies. It would be easy to conclude from the Manhattan plots for the more modestly powered experiments shown in Fig. 1 that the simulated trait has a polygenic basis and that many of the sites with highest −log10(P) scores are true but very modest-effect causative loci. However, through a combination of limited power and genome-wide testing, these apparent polygenic signals are generated through statistical noise alone. This phenomenon, where insufficient statistical power masks a relatively simple genetic architecture, represents a critical interpretive challenge in pool-seq studies.

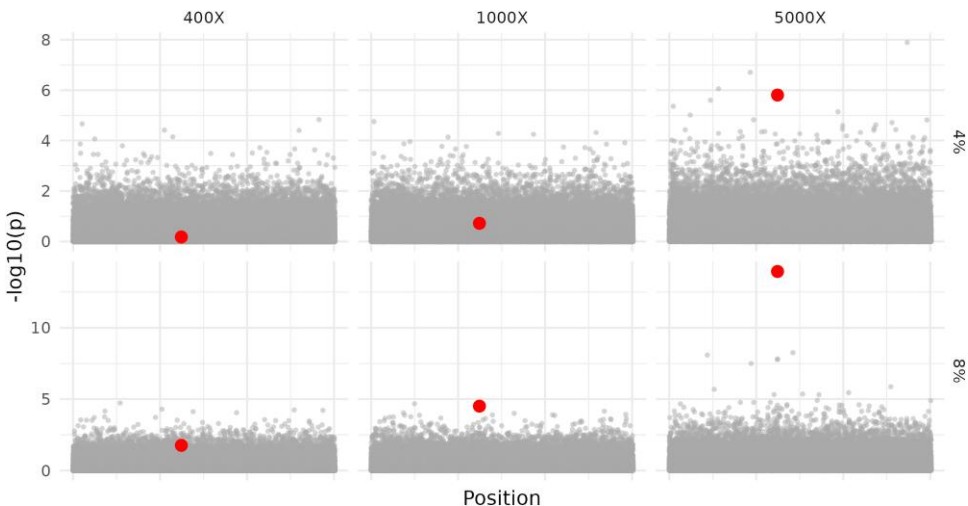

**Fig. 1.** Simulated pool-seq case-control experiments. We simulated 500 haplotypes from a 1 Mb region with typical *D. melanogaster* population genetic parameters. These haplotypes were instantaneously expanded to create 10,000 allele Case vs Control pools, subsequently pool-sequenced to expected coverages of 400×, 1,000×, or 5,000×. The case haplotype pools were sampled conditioning on a single intermediate frequency focal SNP (labeled in red) having a true allele frequency difference between cases or controls of 4% or 8%. We simulate a 1 Mb region here and note a −log10(*P*) threshold of 5 is commonly used in *D. melanogaster* GWAS studies for statistical significance.

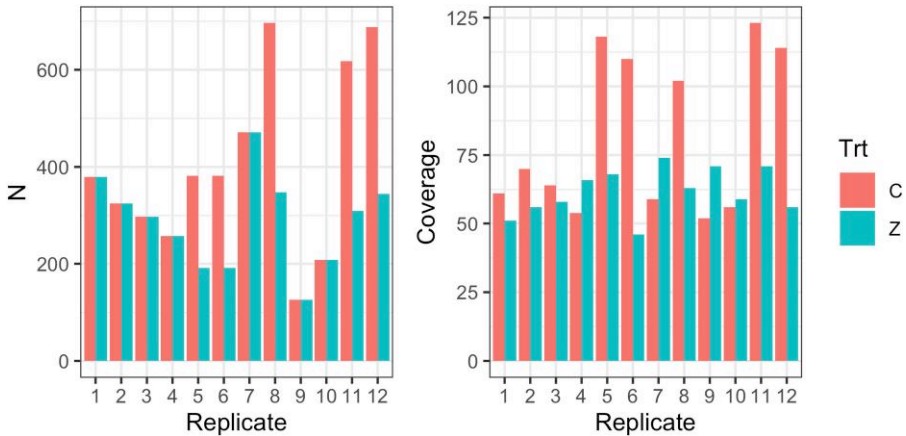

**Fig. 2.** Sampling properties of an X-QTL experiment challenging a *D. melanogaster* synthetic population with zinc chloride (Hanson et al. 2025). The experiment consisted of 12 paired replicates of C̲ontrol vs Z̲inc-selected treatments (Trt). For each treatment and replicate, *N* individuals were sampled to create DNA pools (left panel), and these were subsequently pool-sequenced to the coverage depicted in the right panel. Replicates 5, 6, 8, 11, and 12 had 2 separate, equally sized control pools.

A recently published dataset in *D. melanogaster* allows us to uniquely illustrate the problem in an empirical as opposed to a simulated dataset. Hanson et al. (Hanson et al. 2025) utilized a base population derived by combining 663 DSPR 8-way, known-founder advanced intercross recombinant inbred lines. They collected 0- to 24-h-old embryos and exposed them to either 25 mM zinc chloride or water. Case samples were generated from the ~7% most zinc-resistant adult females that survived the zinc treatment, while controls represent samples of individuals that developed successfully in normal media conditions. Across 12 replicate selection experiments, a total of 4,831 control and 3,448 case individuals were sampled and pooled for sequencing. A total of 739× and 983× sequence coverage for the zinc-treated "case" and water-treated control pools was obtained, respectively (Fig. 2). Ignoring the unique population structure of the base population employed, the experiment is a large case-control GWAS by *Drosophila* standards. Indeed, it is conceivable that the power of the experiment exceeds that of a typical 200–500 inbred-line–based mapping study (Macdonald et al. 2022).

We carried out a GWAS of these data by contrasting directly ascertained allele frequencies between cases and controls, summarizing the experiment with a Manhattan plot (Fig. 3a). Although 642 SNPs exceed a Bonferroni significance threshold, the pattern of these associations is inconsistent with genuine hits. In populations with high linkage disequilibrium, such as the synthetic populations examined here, true causal variants should produce clustered signals spanning multiple linked SNPs over megabase-sized genomic windows. Instead, we observe isolated, dispersed hits throughout the genome, a pattern likely indicative of false positives arising from bioinformatic artifacts (see Supplementary Fig. 1). Specifically, the misalignment of short reads in repetitive or duplicated genomic regions can generate spurious allele frequency differences between cases and controls, an issue that becomes more pronounced with high-coverage sequencing. Critically,

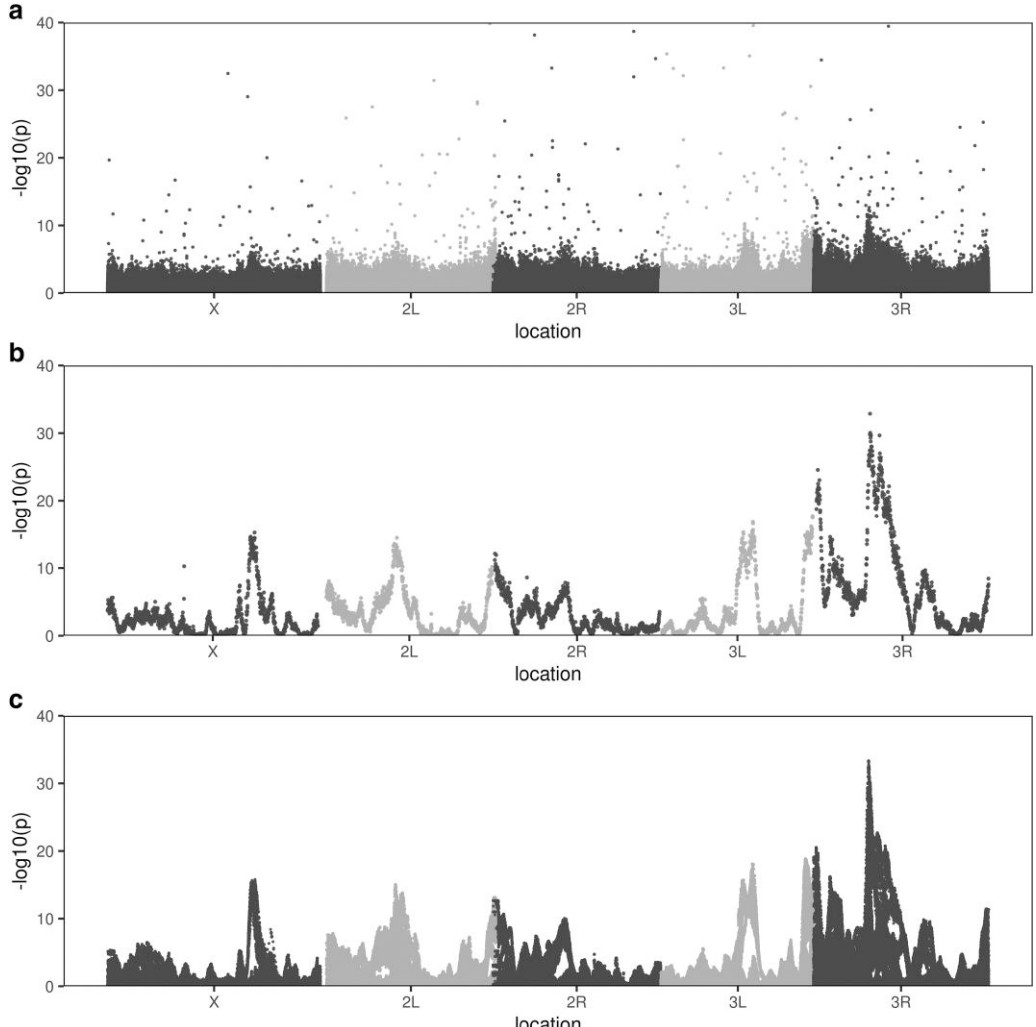

**Fig. 3.** Zinc chloride developmental resistance Manhattan plots associated with 3 different approaches to statistical testing: (a) directly ascertained REF and ALT SNP counts (we do not display a statistical threshold but note a −log10(*P*) threshold of 5 is widely used in *Drosophila* GWAS), (b) imputed founder haplotype counts, and (c) imputed SNP counts. *X*-axes are scaled in genetic as opposed to physical distances. *P*-values are obtained from CMH tests on counts over the 12 replicates of the experiment. For directly ascertained SNPs, the counts used are observed REF vs ALT counts. For the imputation-based tests, "pseudo"-counts are the product of haplotype (or SNP) frequency estimates and twice the number of individuals per DNA pool. These pseudo-counts will inflate −log10(*P*-values) by a constant multiplicative factor.

we observe no genomic regions exhibiting clusters of significant associations that would suggest regions of large effect. Based on these data alone, one could easily conclude that zinc resistance in this population has a highly polygenic genetic architecture.

The dataset of Hanson et al. (Hanson et al. 2025) is unique in that the base population employed is derived from an advanced intercross synthetic population originating from 8 known-founder strains. This experimental design allows the frequency of those 8 founder alleles to be estimated from the pool-seq data throughout the genome using a sliding window haplotype estimator (Long et al. 2011; Kessner et al. 2013; Burke et al. 2014; Tilk et al. 2019; Linder et al. 2020, 2022; Macdonald et al. 2022). Specifically, Hanson et al. estimated founder haplotype frequencies in overlapping 1.5-cM windows across the genome for each sample. These 8 founder haplotype frequencies were then tested for differentiation as a function of genomic location. This known-founder haplotype-based testing strategy underlies the X-QTL (or bulked segregant) analysis approach to mapping QTL (Ehrenreich et al. 2012; Macdonald et al. 2022; Macdonald and Long 2022).

Based on this X-QTL statistical approach, employing the same raw data as the directly ascertained SNP GWAS described above, Hanson et al. concluded that resistance is associated with regions of major effect localized at centimorgan scales. They further identified and validated several major genes in these intervals (including *MTF-1*, a gene central to zinc homeostasis). Figure 3b reproduces the scan of Hanson et al. and clearly shows highly significant peaks of association localized to specific genomic regions. The results of this zinc resistance mapping are consistent with other studies where multiparent population (MPP)-based mapping have identified regions of large effect (King et al. 2012, 2014; Najarro et al. 2015; Everman et al. 2021). They present a stark contrast to the dispersed association pattern observed with directly ascertained SNP-based analysis.

We claim that the primary fundamental reason the haplotype-based test succeeded in detecting major-effect loci, while the direct SNP analysis failed, is simply due to poor power of the latter approach. Despite achieving greater than 700× sequencing coverage per treatment, the directly ascertained SNP approach still

suffers from relatively large binomial sampling errors on the SNP-frequency estimates. In turn, this noise in allele frequency estimation obscures genuine allele frequency differences between pools. In contrast, the haplotype-based approach leverages the critical statistical advantage that haplotype frequency estimates have much smaller errors than directly ascertained SNP frequencies. The error on imputed haplotype frequencies are proportional to ~75% to 100% of the binomial sampling errors on twice the number of individuals per treatment, as opposed to the average short-read coverage (Tilk et al. 2019; Linder et al. 2020). In the zinc chloride experiment the average number of autosomal alleles per treatment $(2 \times 4{,}140 = 8{,}280)$ is roughly 10-fold greater than the average sequence coverage per treatment, and as a result, haplotype frequencies are estimated with much greater certainty than directly ascertained SNP frequencies. The haplotype approach effectively circumvents the coverage limitation of directly ascertaining SNP frequencies. At sequencing coverages much less than $2N$, as is typically the case with pool-seq datasets, imputing founder haplotype frequencies is a "trick" applicable to known-founder populations that can be used to achieve frequency estimation errors "approaching" genotyping every individual.

To illustrate that the difference we observe between the haplotype-based approach (Fig. 3b) and the directly ascertained SNP-based approach (Fig. 3a) is primarily due to sampling error on frequency estimates and not somehow associated with the decision to employ haplotypes or some other detail of X-QTL mapping procedure, we used the haplotype frequency estimates to impute SNP frequencies. That is, we imputed SNP frequencies throughout the genome as the sum of the elementwise product of the founder frequency estimates and known SNP states in the founders at any given location. Although counterintuitive, just like the haplotype frequencies, these imputed SNP frequencies are estimated much more accurately than the directly ascertained SNP counts (Tilk et al. 2019; Linder et al. 2020). We then converted imputed SNP frequencies to pseudo-counts based on the number of individuals in each pool and carried out a third GWAS on the imputed SNP counts (Fig. 3c). The plot demonstrates that when we use the more accurate imputed, as opposed to directly ascertained, SNP allele frequencies, we recover the clear peaks of association seen in the haplotype-based analysis. This confirms that the failure of direct SNP analysis stems from poor power. Furthermore, looking closely at the tests on imputed SNP frequencies (Supplementary Fig. 2), it appears as though one could fit several lines to the data. This pattern represents sets of SNPs that are "in phase" with one another with respect to different founder chromosomes and is expected in multiparent panels derived from a modest number of founders.

## Discussion

The power limitations we demonstrate are not specific to synthetic populations or X-QTL methodology but represent a fundamental challenge for pool-seq approaches. While our analysis employed a synthetic population derived from 8 known founders, the underlying statistical problem that errors on directly ascertained SNP-frequency estimates are large relative to subtle allele frequency differences between pools applies broadly to any pool-seq design. In outbred populations lacking known-founder structure, it is likely that researchers cannot employ the imputation strategy that rescued power in the illustrative example we provide here. Employing pool-seq to routinely detect subtle allele frequency shifts in case-control comparisons using outbred samples

may require 5,000× sequence coverage (and much large sample sizes), roughly equivalent to completely genotyping on the order of 2,500 cases vs control samples. Although there are trait architectures where haplotypes could outperform SNPs, the apparent success of haplotype-based approaches in synthetic populations in this example largely stems not from any inherent superiority of haplotypes over SNPs but simply because haplotype frequencies are estimated more accurately, effectively circumventing the coverage limitations that plague direct SNP ascertainment.

A criticism that can be leveled at the illustrative example presented in this work is that the statistical tests for directly ascertained SNPs use total counts derived from sequence coverage (see Methods), whereas the 2 imputation-based tests use total counts associated with twice the number of individuals in the pool from which DNA is made. Counts associated with the $2N$ individuals in the DNA pools result in higher power, since $2N$ is more than ten times the sequence coverage per pool. However, the total sequence coverage is the true count associated with directly ascertained SNPs, whereas the $2N$ associated with imputation-based tests incorrectly assumes imputation is equivalent to perfectly genotyping the $N$ individuals in each pool. Imputation is simply not that efficient. But the decision to multiply frequencies by $2N$ to obtain counts "only" impacts the statistical test as a multiplicative constant throughout the genome. As a result, the imputation-based Manhattan plots of Fig. 3b and c have an "inflated" Y-axis scale, but this inflation constant does not contribute to the noise of the directly ascertained GWAS.

A second criticism is that an MPP is not appropriate for a GWAS-type analysis, since a synthetic population derived from an 8-way cross is not the type of population typically employed in a GWAS study (generally a sample of natural chromosomes lacking a known set of founders). This is reflected in Fig. 3 where for the 2 imputation-based scans the "hits" extend over centimorgan-sized intervals, a much larger region than the localization signal expected of a GWAS. However, this difference does not detract from the value of our illustration. We observe that directly ascertained SNP frequencies at coverages approaching 700× per sample are insufficient to detect the real, but subtle frequency shifts apparent in this dataset, a result independent of the genetic details of the population employed. If anything, the problem is more dire in a population harboring lower levels of linkage disequilibrium as a function of distance. In such a case, one is trying to detect the signal associated with only a handful of SNPs in LD with a causative factor, and a lower genome-wide $P$-value threshold is likely required to control for false positives.

In *D. melanogaster*, we are aware of 4 traits have been dissected using both an MPP-panel based mapping approach using the DSPR (King et al. 2012) and a GWAS-based analysis using collection of inbred lines called the *Drosophila* Genetic Reference panel (DGRP; Mackay et al. 2012; Huang et al. 2014) where the genetic architecture appears complex: resistance to caffeine (Najarro et al. 2015), copper toxicity (Everman et al. 2021, 2023), boric acid resistance (Najarro et al. 2017), and starvation stress (Everman et al. 2019). In all 4 cases DSPR mapping-based approaches have successfully identified loci of large effect, while in contrast DGRP association-based approaches have concluded that trait variation is highly polygenic and due to dozens to hundreds of loci of subtle effect spread throughout the genome. Notably the association hits are subtle enough that they rely on a suggestive threshold for significance, often a $-\log_{10}(P\text{-value}) > 5$. Our observations with respect to pool-seq case-control designs suggest that power deficits can lead to incorrect assumptions about the polygenicity of complex traits, and we speculate that a similar pattern may explain the

contradictory results observed in these inbred line studies. The highly polygenic signal in these 4 association studies is consistent with DGRP-based GWAS outcomes more generally (Mackay and Huang 2018), as well as those in other *Drosophila* GWAS-type studies employing individual-based genotyping (Pallares et al. 2023). Overall, in *Drosophila* the genetic basis of trait variation dissected using GWAS-type approaches appears highly polygenic, while in contrast traits dissected using QTL-mapping–type approaches appear more oligogenic, and this remains true even when holding the trait constant. It is possible that the apparent contradiction between polygenicity vs handfuls of more major-effect loci is driven by power.

## Data and methods availability

The raw data, methods for calling SNPs, and haplotyping calling are described in Hanson et al. (2025). DSPR RILs are available from the BDSC, raw FASTQ files are in the NCBI SRA under accession number PRJNA1127662, and all code to replicate the analyses is available on GitHub (https://github.com/Hanson19/Zinc-X-QTL). The analysis code and code to reproduce the figures of the present work are hosted on Github (https://github.com/tdlong/Compare_3_GWAS.git).

Supplemental material available at *GENETICS* online.

## Funding

This work was supported by NIH award R01-OD034064 (to SJM and ADL).

## Conflicts of interest

None declared.

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

Editor: P. Wittkopp