## [Peer Review File · Genetics]

The Illusion of Polygenicity in Poolseq studies: Insufficient Power Can Mask Simple Genetic Architectures

Anthony Long, Katherine Hanson, and Stuart Macdonald

NOTE: The reviews and decision letters are unedited and appear as submitted by the reviewers.

In extremely rare instances and as determined by a Senior Editor or the EIC, portions of a review may be redacted. If a review is signed, the reviewer has agreed to no longer remain anonymous.

The review history appears in chronological order.

Review Timeline:

Submission Date:	2025-07-22
Editorial Decision:	2025-08-12
Resubmission Received:	2025-10-03
Editorial Decision:	2025-11-05
Resubmission Received:	2025-12-29
Editorial Decision:	2026-01-26
Revision Received:	2026-02-12
Accepted:	2026-02-19

August 12, 2025

GENETICS-2025-308416

The Illusion of Polygenicity in Poolseq studies: Insufficient Power Can Mask Simple Genetic Architectures

Dear Dr. Long:

Two experts in the field have reviewed your manuscript, and I have read it as well. We all appreciate the importance of trying to integrate findings from QTL mapping and GWAS. Unfortunately, as you will see below, the reviewers found the work to fall short of that goal, to insufficiently incorporate prior work on this topic, and to be too *Drosophila*-centric at times.

While your manuscript is not currently acceptable for publication in GENETICS, we would welcome a substantially revised manuscript. Both reviewers have comments and concerns to be addressed in a revised manuscript. You can read their reviews at the end of this email. Please note that a successful revision of this work will need to make the manuscript accessible for and of interest to a broader readership. Transferring the manuscript to G3 might also be an option to consider, and reviewer 2 suggests considering incorporating points made in this paper with another paper currently on BioRxiv as an alternative to publishing it separately.

We look forward to receiving your revised manuscript, should you choose to submit one. Please let the editorial office know approximately how long you expect to need for revisions.

Upon resubmission, please include:

1. A clean version of your manuscript;
2. A marked version of your manuscript in which you highlight significant revisions carried out in response to the major points raised by the editor/reviewers (track changes is acceptable if preferred);
3. A detailed response to the editor's/reviewers' feedback and to the concerns listed above. Please reference line numbers in this response to aid the editor and reviewers.

Your paper will be sent back out for review.

Additionally, please ensure that your resubmission is formatted for GENETICS
<https://academic.oup.com/genetics/pages/general-instructions>

Follow this link to submit the revised manuscript: Link Not Available

Sincerely,

Patricia Wittkopp
Associate Editor
GENETICS

Approved by:
Mario Calus
Senior Editor
GENETICS

Reviewer #1 :

Despite the introduction and abstract, I don't think this paper is about QTL vs GWAS. Nor is it about power of QTL vs GWAS at least not directly. The rest of the paper (results and discussion with the very brief methods) is quite informal and reads as if it is directed to me, someone very familiar with the *Drosophila* complex-trait mapping literature, who knows the populations and the ins and outs of the discussions over time. There is a disconnect here in the set-up expectation of the target audience between the abstract/introduction P1 and the rest of the paper.

P1 Introduction:

The QTL vs GWAS set up in this paragraph is a very 'loose' description and it is not an effective entree into the results presented here. When I think about the general topic of experimental design for identifying loci associated with complex trait phenotypes, I think about the target population (natural or breeding) and then the number of loci and the number of alleles per locus. In the

end you are talking about 'allele tracking' regardless of whether these are within or between loci. When you are discussing a GWAS of a large natural population, the number of alleles per locus can be quite large and the effect sizes per allele vary as can the allelic combinations. QTL studies, typically though not necessarily have a maximum number of alleles per locus (2 at the start then with MPP larger numbers). The attempt to 'synchronize' results only makes sense for the same trait in the same species -and the final sentence here makes little sense ...

There is a fair bit of work on integrating association and mapping approaches (much of it published in GENETICS) and a robust discussion of the issues around alleles/populations as well as statistical tests, power and sample size in the plant genetics literature, all of which is ignored here.

P2 introduction

This is a very informal summary that presumes familiarity with Drosophila genetics.

I think the authors raise a number of issues and these issues are 1) not dealt with in a precise manner ... 2) not a subject of the results presented here... so I am confused about why they are raised here . For example:

- the number of tests does not contribute to the p-value estimate as this sentence seems to suggest (likely inadvertently). The p-value resolution (how many decimal places can be estimated) is a function of the sample size (line number) and allele frequency, not the number of SNPs tested. The threshold used to decide significance is a function of the number of tests for a FWER approach but not exactly that for a FDR.

- the bonferroni threshold is has been largely abandoned as a standard (rightly in my opinion) and the authors are correct there is this prescedent for a 'suggestive' threshold of 10^{-5} which was established in the 2012 paper that introduced the DGRP. This is tied to "the smallest" p-value size estimable from the DGRP given the number of lines or put another way what are the effective number of independent tests and the allele frequency that can be detected with this panel. However just because this is the 'smallest' estimable doesn't necessarily mean the type I error rate is low... and while the Drosophila community has largely ignored the type I error in these studies, the FDR is important and ignoring it in GWAS approaches is not generally "allowed" and indeed there is literature on FDR in mapping. one possible explanation for finding 'highly polygenic signal' is just type I error...

The elephant in the room here is the FDR, though I understand that perhaps the authors wish to avoid a discussion of this... but there is really alot of very nice work around the use of the FDR in both GWAS and QTL (e.g. B&Y 2005 Genetics) and this might actually clarify some of the apparent 'discrepancies' ...

The statement in lines 94-96 appears overly general for the handful of traits studied with more than 1 experimental design in drosophila genetics...

P3 introduction

I think this is meant to be an introduction to pool-seq and the paper would benefit from a cartoon figure describing the population/pool-seq approach that is used here.

Line 115/116 the GWAS is uncovering a polygenic architecture and the QTL mapping of the same dataset is not... BUT i don't actually think that is what the results show...

you are using the same test statistic in all three analyses, and indeed the same data

I think the simulation is about the accuracy of SNP frequency estimates and the importance of leveraging the haplotypes, thereby reducing the SNP frequency error. There is an example dataset which shows (roughly) the result the simulation is meant to explain.

I think the results are related to the topic: how haplotypes help reduce error in SNP genotyping. browning 2009 (PMID: 18850115) and Howie Donnelly and marchini 2009 (PMID: 19543373), they both have a similar end result and I think there are a number of other subsequent papers on this topic. I think these were the first but I could be wrong here and there may be others.

Here the 'twist' is the pool-seq and so the haplotypes in the starting population are known but the frequency of the haplotypes in the different designs is a bit unclear. The actual haplotype estimation is, I think, non-trivial. Here perhaps it is easy in this example? But this is worth a little background/literature franssen Barton, Schlotterer 2017, Barghi and Schlotterer 2019, Otte and Schlotterer 2021, Tilk et. al. 2019

then how the precision in the frequency estimates of the SNPs in the tail fractions is improved by leveraging the haplotypes. This is the 'new' information and I think it makes sense and is an interesting result. This may be true for all pool-seq in Drosophila even if the tail selection is different than in an X-QTL. There are other Drosophila experiments that use pool-seq approaches and I think these also benefit from the haplotype estimation (see above)

Results/Disussion

Paragraph 1 is an informal description of the X-QTL study design. While the population studied is large the number of alleles per

locus is a maximum of 8. We also have no information about the starting size of the haplotype blocks and based on what is described here I don't have a good sense of the recombination in the population. The assertion line 142 about the clarity of the results (in the absence of a threshold or any estimate of the type 1 error either FDR or FWER) seems optimistic...

paragraph 2/3-- i am not sure it makes sense to argue that you can test individual snps as if this is a GWAS on a "natural population". It is a "case/control" study. That is in effect the tail comparison X-QTL argument and why the X-QTL "works". The population is what it is by experimental design, pretending you don't know the population structure, does not change the underlying population structure nor does it prove that a different underlying population structure would produce the same result.

186-221 I think the result is line 209 (italics). This is the result. The type 1 error rate, and a threshold should be formally calculated. A discussion of how error rates in allele frequencies can be estimated and also drive false positives can (and should) be more explicitly and precisely made.

The argument seems to be: when you 'look' at the three pictures, if the middle one were a GWAS of a population that had a different structure than it really has, you might say something different about the number of significant SNPs. This argument does not make sense to me.

What does make sense to me is that in this set -up, because of this population structure, you can directly estimate the error rate in the SNP frequency estimates from pool-seq and these are high. This has implications for pool-seq. I am not convinced this has implications for the DGRP where the issue is not SNP sequencing error, but the limited number of lines.

methods

line 296-307 how big was the population? what was the phenotype exactly? I don't want to have to go to hanson 2025 to read this how were the number of adult females pooled per replicate determined? (I would explain the issue raised in lines 322-325 at the start.

line 310-311 delete this instead focus on what you did do here and why. I would also describe the CMH after the description of the SNP calling

Line 314-325 I think starting with counting ref/alt at the SNP what was used the number of ref reads and the number of alt reads? After this I don't understand what you did to 1) get the haplotypes and then 2) impute the SNPs. Maybe less is better here and just reference where you describe this in detail and point to the data file that has the raw and imputed data I would appreciate knowing how many SNP positions there are and what % differ between direct counts and imputation 1 and 2.

Code should have a permanent doi associated with it - it can be supplement here or a snapshot of the github can be archived.-

Figure 1: What is "coverage" here ?

Figure 2: This is the same data analyzed three ways? These three are a) imputed haplotype b) at each snp based on ref/alt (formula?) and c) imputed SNP this I think correposnds to the description lines 314-325. I am struggling to follow the words. I am also struggling to see what I should take away from this- these are the same? different? Maybe a plot of the difference in the test statistic? Maybe this plot should have a fourth panel of a spline (or other smooth curve estimator) from each of the three methods overlaid?

figure 3: B/C) The Y axis should be the rate not the log10 of the rate... and with simulations you should be able to estimate the variance in the rate - I suspect the weird blips in the purple and teal in panel B , and the strange 0 values obtained by dividing by N would go away if you repeated the simulation and then took the mean. There is no reason to do just one instance of a simulation in such a simple scenario.

What do you mean by coverage? how exactly is the simulation done? is it generating X reads at random to produce an average of a number of reads per SNP? I find the two sentences lines 327-329 opaque. Since you invoke the r function rbinom, I assume that you are using a binomial distribution but it would be better to say this more formally, what allele frequency ?

minor

line
66 genetic dissection, #68 dissection. I know what you mean but I am not sure dissection is the best verb.

line #146 highly significant does not exist, there is a threshold and the result is or is not significant. What I think you mean is that you have small p-values.

Reviewer #3 :

I assume this is a Brief Investigation (3000 words max), but this was not clear.

Overall I found this to be an interesting paper and fun to read, but I am unclear as to its intended take home message.

This paper requires one to understand the X-QTL methodology described in Hansen et al 2025, currently a BioRxiv paper. It would seem more logical to combine the papers into one, since the current paper is essentially a demonstration that a naïve SNP-calling approach does not work as well as a haplotype approach, a result which is certainly worth sharing, but probably makes more sense as part of a larger publication. My take home was that the X-QTL approach solves the problem, so was not sure why it needs to be a separate paper. But perhaps I have missed something.

The causes of the differences between a SNP-based analyse (Figure 1B) seem to be partly to do with bioinformatics choices, and are fixable as shown in Fig 1C. The question is why the logP values of the "true" signal in Fig 1B track those in Fig 1A,C but at reduced level - is this truly a loss of power or is it because the genome wide thresholds are correspondingly lower?

Specific points

Abstract mentions X-QTL without defining it

L 141-143 - you could estimate the genome wide thresholds for significance in Fig 1A, B, C by permuting the treatment/control labels in each of the 12 replicates, resulting in 2^{12} possible relabellings. Recording the genome wide max score from a sample of say 10000 permutations would give you a sense of the threshold.

P5 lines 159-169 - paragraph contains some speculative assertions eg "At these coverages, a case/control design may have greater power than a ~200 inbred line-based GWAS...." - perhaps these should be in the discussion, and better justified.

P5 L 172 onwards. I agree with the authors that the miasma of isolated high SNP -based associations in Fig 2B are likely artefacts, possibly due to read alignment problems and base-calling errors. Did the authors investigate this further? First, are the allele frequencies of these SNPs markedly lower than the average (and in general what is the allele frequency spectrum like - one might expect them to be very low frequency high effect associations and do the pooled allele frequencies match those expected?) Second, can the reference genome annotations be used to identify and mask out different classes of sequence (eg mobile elements, intergenic, repeats, etc) - if attention is restricted just to coding sequence, what happens?

I agree that the Figure 2C (using imputed SNPs via haplotypes) closely resembles Figure 2A. In a sense this is to be expected - provided the chromosomes of the mapping population are well approximated by mosaics of known haplotypes (which should be the case for this population) then imputation should be superior to directly ascertained SNPs. What happens in Figure 2C to the allele frequencies of the "associated" SNPs in Figure 2B? - one assumes these are unstable and therefore markedly different. The separation of imputed SNPs into those with high vs low significance is to be expected - this depends on whether they are tagging the right split of haplotypes. I would have expected there to be some QTL where no imputed SNP gave the right split (ie the effect is irreducibly haplotypic and not biallelic).

P6 Line 223 onwards. I am puzzled by the result that when sequencing coverage is higher than the number of individuals in the pool then the false positive rate increases. Can the authors provide a better explanation for this?

Response to Reviewers:

The two reviewers identified a major problem with the paper. We did a terrible job of really stating what we were trying to do. What was left was a Brief Investigation that failed. As opposed to trying to rework the paper and respond to the reviewers point-by-point we completely rewrote the paper from scratch (except for one paragraph that was part of the introduction that is now part of the discussion), including the figures. Below we try to explain why we think we failed, the big changes, and address some themes touched on by the reviewers, and how hopefully they are addressed in the rewritten manuscript.

It is our hope the reviewers can have another look at the paper, perhaps see more clearly what we are trying to do, and then re-evaluate the work within this new framework. We apologize if this creates additional work, but feel it reflects our taking their initial concerns very seriously.

What we are trying to do in the paper is to address a pattern we have seen many times in the literature (and in talks) where labs do something akin to a case/control genetic mapping study using pool-seq, and then interpret their results via a Manhattan plot of SNP-by-SNP P-values. A common way this has played out in *Drosophila* (where the best examples of this practice exist) is a group creates a DNA pool from phenotypically-extreme flies obtained from some treatment (along with a control pool), pool-sequences that DNA to perhaps 50-200X, and then creates a Manhattan plot contrasting SNP allele frequencies in the case pool versus the control pool. What is often observed is no clear pattern of unambiguously significant hits, which is then interpreted as the trait showing a strong polygenic basis. What we have tried to show in our paper is that these studies are extremely underpowered, and that even if there are genes of large effect, they can easily go undetected. As a result it can be dangerous to interpret a lack of such a signal in a Manhattan plot.

The key empirical finding is that even at 700X sequence coverage per pool, which is far higher than in typical studies, meaningful regions of large effect are undetectable in a direct SNP-by-SNP GWAS, but are clearly identified when haplotype information was leveraged.

We simulate exactly this scenario (as Figure 1), but this is not so novel. It is pretty easy to show that the power to detect an allele frequency difference of 5% at a p-value of 10^{-5} given binomial samples at 100X coverage per pool is essentially zero, yet that has not stopped investigators from drawing conclusions from Manhattan plots representing exactly this experiment.

In exploring the Hanson et al dataset (which is now published in *Genetics* - the status of that paper should have been better articulated in the initial cover letter), we realized that it was a pretty large pool-seq case control experiment, larger actually than anything published in flies. We carried out a case/control comparison on the directly ascertained SNPs, and saw no clear hits. But what was interesting about this dataset is that when we exploit the fact that the genetic design is a multi-parental panel, and we impute founder haplotypes, there are clearly regions of large effect as shown in the Hanson paper (and in Figure 3B in the current work). We go on to show that the failure of the directly ascertained SNP-based GWAS experiment is due to the fact that even at 700X sequence coverage in cases vs controls, there is insufficient power to detect meaningful regions of large effect. This dataset allows for a clear empirical example of why one cannot conclude much about genetic architecture from a pool-seq, case-control genetic mapping study that employs only *modest* sequence coverage – where "*modest*" is much greater than anything close to what investigators are doing.

To be completely clear: this paper addresses only pool-seq case/control designs, not GWAS on individual genotyped lines like the DGRP. We have now clarified that our analysis only speaks directly to doing GWAS using pool-seq on case versus control samples (a common fly design). Although in the discussion we point out that GWAS using DGRP have tended to detect different architectures than MPP-based genetic QTL mapping for the same trait, and power may have something to do with that observation, but that remains speculative. The reviewers did not call us out on this, but we may have over-reached in the original paper and muddled GWAS on inbred panels, with GWAS from pooled DNA samples.

The original manuscript completely failed to convey the above message. This was pointed out by both reviewers in their opening sentences (and highlighted in RED in their reviews)! Our failure to clearly articulate our message made the paper really difficult to review and comment on, it was unfair to the reviewers. We took the reviews to heart and re-wrote the manuscript. Below we attempt to identify some themes touched on by the reviewers and how we think they are addressed in the rewrite.

Reviewer 1 points out that there is considerable literature on integrating association and mapping. We are aware of that literature, but that is not the point here (and would not really apply given the small number of founders in the DSPR), as our goal is merely to illustrate that one cannot correctly interpret Manhattan plots from poorly powered pool-seq case-control studies.

Reviewer 1 also discussed p-values and FDR. We agree that FDR is important in GWAS interpretation generally. But this paper is not about establishing significance thresholds. Our point is more fundamental: that investigators cannot conclude a trait is highly polygenic simply because they observe no clear hits, regardless of the statistical threshold used, when a study is underpowered to detect even large effects. We are not weighing in on the appropriateness of a threshold of 10[-5] in the fly literature. We are more pointing out the opposite, that just because you don't have hits at some modest FDR, doesn't mean the trait is highly polygenic. We are not saying that investigators should leverage haplotypes (an advantage of X-QTL but this requires a special population). What we are saying is that for the Hanson et al dataset haplotypes seems to show regions of pretty large effect (including some smoking gun candidate genes), yet the GWAS fails to detect these regions, despite being a better GWAS in terms of sequence coverage than is typical in the literature.

Reviewer 3 thinks one needs to understand the X-QTL methodology and perhaps this paper should have been rolled into the Hanson et al paper. We think this paper makes a separate point (and the Hanson et al paper was already accepted pending revision when this paper was submitted). We hope the rewrite makes clear that this paper is not about Zinc chloride resistance, and that you can understand this paper X-QTL details aside. Our key observation is that the Hanson et al dataset is a really big case control study by Drosophila standards (using 700X sequence coverage per treatment) and the directly ascertained GWAS approach fails to detect major genes that the X-QTL approach discovers. We are not comparing traits or studies here – this is the same dataset. What we are trying to show is that even at 700X sequence coverage, fairly large allele frequency differences between cases and controls are not easily detected. X-QTL exploits haplotype information to estimate allele frequencies more precisely than binomial sampling errors on coverage. We presume (and indeed try to show) that if we had sequenced to higher coverage (akin to genotyping something like 3000 cases vs controls like in human genetics) a different genetic architecture would have been apparent. X-QTL highlights the problem of attempting to draw inference about genetic architecture from Manhattan plots even at 700X sequence coverage per pool.

We are not trying to critique the reviewer's comments here. The way they saw the paper, given how it was written, was a problem with the paper, not the reviewers. If we are claiming above that they missed the point, it is because we failed to make it.

In summary, we believe the complete rewrite addresses our fundamental communication failures of the original manuscript, and hope the reviewers can now evaluate our contribution within the proper framework: demonstrating that current pool-seq case/control studies are severely underpowered, even at coverage levels far exceeding current practice., And that such investigations are making conclusions about trait genetic architecture from negative results that are potentially misleading.

original reviews
#####

Reviewer #1 :

Despite the introduction and abstract, I don't think this paper is about QTL vs GWAS. Nor is it about power of QTL vs GWAS at least not directly. The rest of the paper (results and discussion with the very brief methods) is quite informal

and reads as if it is directed to me, someone very familiar with the Drosophila complex-trait mapping literature, who knows the populations and the ins and outs of the discussions over time. There is a disconnect here in the set-up expectation of the target audience between the abstract/introduction P1 and the rest of the paper.

P1 Introduction:

The QTL vs GWAS set up in this paragraph is a very 'loose' description and it is not an effective entry into the results presented here. When I think about the general topic of experimental design for identifying loci associated with complex trait phenotypes, I think about the target population (natural or breeding) and then the number of loci and the number of alleles per locus. In the end you are talking about 'allele tracking' regardless of whether these are within or between loci. When you are discussing a GWAS of a large natural population, the number of alleles per locus can be quite large and the effect sizes per allele vary as can the allelic combinations. QTL studies, typically though not necessarily have a maximum number of alleles per locus (2 at the start then with MPP larger numbers). The attempt to 'synchronize' results only makes sense for the same trait in the same species -and the final sentence here makes little sense ...

There is a fair bit of work on integrating association and mapping approaches (much of it published in GENETICS) and a robust discussion of the issues around alleles/populations as well as statistical tests, power and sample size in the plant genetics literature, all of which is ignored here.

P2 introduction

This is a very informal summary that presumes familiarity with Drosophila genetics.

I think the authors raise a number of issues and these issues are 1) not dealt with in a precise manner ... 2) not a subject of the results presented here... so I am confused about why they are raised here . For example:

- the number of tests does not contribute to the p-value estimate as this sentence seems to suggest (likely inadvertently). The p-value resolution (how many decimal places can be estimated) is a function of the sample size (line number) and allele frequency, not the number of SNPs tested. The threshold used to decide significance is a function of the number of tests for a FWER approach but not exactly that for a FDR.
- the bonferroni threshold has been largely abandoned as a standard (rightly in my opinion) and the authors are correct there is this precedent for a 'suggestive' threshold of 10^{-5} which was established in the 2012 paper that introduced the DGRP. This is tied to "the smallest" p-value size estimable from the DGRP given the number of lines or put another way what are the effective number of independent tests and the allele frequency that can be detected with this panel. However just because this is the 'smallest' estimable doesn't necessarily mean the type I error rate is low... and while the Drosophila community has largely ignored the type I error in these studies, the FDR is important and ignoring it in GWAS approaches is not generally "allowed" and indeed there is literature on FDR in mapping. one possible explanation for finding 'highly polygenic signal' is just type I error...

The elephant in the room here is the FDR, though I understand that perhaps the authors wish to avoid a discussion of this... but there is really a lot of very nice work around the use of the FDR in both GWAS and QTL (e.g. B&Y 2005 Genetics) and this might actually clarify some of the apparent 'discrepancies' ...

The statement in lines 94-96 appears overly general for the handful of traits studied with more than 1 experimental design in drosophila genetics...

P3 introduction

I think this is meant to be an introduction to pool-seq and the paper would benefit from a cartoon figure describing the population/pool-seq approach that is used here.

Line 115/116 the GWAS is uncovering a polygenic architecture and the QTL mapping of the same dataset is not... BUT I don't actually think that is what the results show...

you are using the same test statistic in all three analyses, and indeed the same data

I think the simulation is about the accuracy of SNP frequency estimates and the importance of leveraging the haplotypes, thereby reducing the SNP frequency error. There is an example dataset which shows (roughly) the result the simulation is meant to explain.

I think the results are related to the topic: how haplotypes help reduce error in SNP genotyping. Browning 2009 (PMID: 18850115) and Howie Donnelly and Marchini 2009 (PMID: 19543373), they both have a similar end result and I think there are a number of other subsequent papers on this topic. I think these were the first but I could be wrong here and there may be others.

Here the 'twist' is the pool-seq and so the haplotypes in the starting population are known but the frequency of the haplotypes in the different designs is a bit unclear. The actual haplotype estimation is, I think, non-trivial. Here perhaps it is easy in this example? But this is worth a little background/literature: Franssen Barton, Schlotterer 2017, Barghi and Schlotterer 2019, Otte and Schlotterer 2021, Tilk et al. 2019

then how the precision in the frequency estimates of the SNPs in the tail fractions is improved by leveraging the haplotypes. This is the 'new' information and I think it makes sense and is an interesting result. This may be true for all pool-seq in *Drosophila* even if the tail selection is different than in an X-QTL. There are other *Drosophila* experiments that use pool-seq approaches and I think these also benefit from the haplotype estimation (see above)

Results/Disussion

Paragraph 1 is an informal description of the X-QTL study design. While the population studied is large the number of alleles per locus is a maximum of 8. We also have no information about the starting size of the haplotype blocks and based on what is described here I don't have a good sense of the recombination in the population. The assertion line 142 about the clarity of the results (in the absence of a threshold or any estimate of the type 1 error either FDR or FWER) seems optimistic...

paragraph 2/3-- i am not sure it makes sense to argue that you can test individual snps as if this is a GWAS on a "natural population". It is a "case/control" study. That is in effect the tail comparison X-QTL argument and why the X-QTL "works". The population is what it is by experimental design, pretending you don't know the population structure, does not change the underlying population structure nor does it prove that a different underlying population structure would produce the same result.

186-221 I think the result is line 209 (italics). This is the result. The type 1 error rate, and a threshold should be formally calculated. A discussion of how error rates in allele frequencies can be estimated and also drive false positives can (and should) be more explicitly and precisely made.

The argument seems to be: when you 'look' at the three pictures, if the middle one were a GWAS of a population that had a different structure than it really has, you might say something different about the number of significant SNPs. This argument does not make sense to me.

What does make sense to me is that in this set-up, because of this population structure, you can directly estimate the error rate in the SNP frequency estimates from pool-seq and these are high. This has implications for pool-seq. I am not convinced this has implications for the DGRP where the issue is not SNP sequencing error, but the limited number of lines.

methods

line 296-307 how big was the population? what was the phenotype exactly? I don't want to have to go to hanson 2025 to read this how were the number of adult females pooled per replicate determined? (I would explain the issue raised in lines 322-325 at the start.

line 310-311 delete this instead focus on what you did do here and why. I would also describe the CMH after the description of the SNP calling

Line 314-325 I think starting with counting ref/alt at the SNP what was used the number of ref reads and the number of alt reads? After this I don't understand what you did to 1) get the haplotypes and then 2) impute the SNPs. Maybe less is better here and just reference where you describe this in detail and point to the data file that has the raw and imputed data I would appreciate knowing how many SNP positions there are and what % differ between direct counts and imputation 1 and 2.

Code should have a permanent doi associated with it - it can be supplement here or a snapshot of the github can be archived.-

Figure 1: What is "coverage" here ?

Figure 2: This is the same data analyzed three ways? These three are a) imputed haplotype b) at each snp based on ref/alt (formula?) and c) imputed SNP this I think corresponds to the description lines 314-325. I am struggling to follow the words. I am also struggling to see what I should take away from this- these are the same? different? Maybe a plot of the difference in the test statistic? Maybe this plot should have a fourth panel of a spline (or other smooth curve estimator) from each of the three methods overlaid?

figure 3: B/C) The Y axis should be the rate not the log10 of the rate... and with simulations you should be able to estimate the variance in the rate - I suspect the weird blips in the purple and teal in panel B , and the strange 0 values obtained by dividing by N would go away if you repeated the simulation and then took the mean. There is no reason to do just one instance of a simulation in such a simple scenario.

What do you mean by coverage? how exactly is the simulation done? is it generating X reads at random to produce an average of a number of reads per SNP? I find the two sentences lines 327-329 opaque. Since you invoke the r function rbinom, I assume that you are using a binomial distribution but it would be better to say this more formally, what allele frequency ?

minor

line

66 genetic dissection, #68 dissection. I know what you mean but I am not sure dissection is the best verb.

line #146 highly significant does not exist, there is a threshold and the result is or is not significant. What I think you mean is that you have small p-values.

Reviewer #3 :

I assume this is a Brief Investigation (3000 words max), but this was not clear.

Overall I found this to be an interesting paper and fun to read, but I am unclear as to its intended take home message.

This paper requires one to understand the X-QTL methodology described in Hansen et al 2025, currently a BioRxiv paper. It would seem more logical to combine the papers into one, since the current paper is essentially a demonstration that a naïve SNP-calling approach does not work as well as a haplotype approach, a result which is certainly worth sharing, but probably makes more sense as part of a larger publication. My take home was that the X-QTL approach solves the problem, so was not sure why it needs to be a separate paper. But perhaps I have missed something.

The causes of the differences between a SNP-based analyse (Figure 1B) seem to be partly to do with bioinformatics choices, and are fixable as shown in Fig 1C. The question is why the logP values of the "true" signal in Fig 1B track those in Fig 1A,C but at reduced level - is this truly a loss of power or is it because the genome wide thresholds are correspondingly lower?

Specific points

Abstract mentions X-QTL without defining it

L 141-143 - you could estimate the genome wide thresholds for significance in Fig 1A, B, C by permuting the treatment/control labels in each of the 12 replicates, resulting in 2^{12} possible relabellings. Recording the genome wide max score from a sample of say 10000 permutations would give you a sense of the threshold.

P5 lines 159-169 - paragraph contains some speculative assertions eg "At these coverages, a case/control design may have greater power than a ~200 inbred line-based GWAS...." - perhaps these should be in the discussion, and better justified.

P5 L 172 onwards. I agree with the authors that the miasma of isolated high SNP -based associations in Fig 2B are likely artefacts, possibly due to read alignment problems and base-calling errors. Did the authors investigate this further? First, are the allele frequencies of these SNPs markedly lower than the average (and in general what is the allele frequency spectrum like - one might expect them to be very low frequency high effect associations and do the pooled allele frequencies match those expected?) Second, can the reference genome annotations be used to identify and mask out different classes of sequence (eg mobile elements, intergenic, repeats, etc) - if attention is restricted just to coding sequence, what happens?

I agree that the Figure 2C (using imputed SNPs via haplotypes) closely resembles Figure 2A. In a sense this is to be expected - provided the chromosomes of the mapping population are well approximated by mosaics of known haplotypes (which should be the case for this population) then imputation should be superior to directly ascertained SNPs. What happens in Figure 2C to the allele frequencies of the "associated" SNPs in Figure 2B? - one assumes these are unstable and therefore markedly different. The separation of imputed SNPs into those with high vs low significance is to be expected - this depends on whether they are tagging the right split of haplotypes. I would have expected there to be some QTL where no imputed SNP gave the right split (ie the effect is irreducibly haplotypic and not biallelic).

P6 Line 223 onwards. I am puzzled by the result that when sequencing coverage is higher than the number of individuals in the pool then the false positive rate increases. Can the authors provide a better explanation for this?

November 5, 2025

GENETICS-2025-308668

The Illusion of Polygenicity in Poolseq studies: Insufficient Power Can Mask Simple Genetic Architectures

Dear Dr. Long:

Two experts in the field have reviewed your revised manuscript, and I have read it as well. We all agree it is much more clear, but some important questions require further attention, as described below. While your manuscript is not currently acceptable for publication in GENETICS, we would welcome a revised manuscript.

We look forward to receiving your revised manuscript. Please let the editorial office know approximately how long you expect to need for revisions.

Upon resubmission, please include:

1. A clean version of your manuscript;
2. A marked version of your manuscript in which you highlight significant revisions carried out in response to the major points raised by the editor/reviewers (track changes is acceptable if preferred);
3. A detailed response to the editor's/reviewers' feedback and to the concerns listed above. Please reference line numbers in this response to aid the editor and reviewers.

Your paper will likely be sent back out for review.

Additionally, please ensure that your resubmission is formatted for GENETICS

<https://academic.oup.com/genetics/pages/general-instructions>

Follow this link to submit the revised manuscript: Link Not Available

Sincerely,

Patricia Wittkopp
Associate Editor
GENETICS

Approved by:
Mario Calus
Senior Editor
GENETICS

Reviewer #1 :

I think your argument is this: directly ascertaining SNPs and using the SNP frequency from pool-seq underestimates the sampling variance (it is the wrong test statistic). When the sampling variance is properly accounted for- either by adopting a haplotype test based on windows, or by correcting the SNP estimates then the false positive rate is controlled.

That is I think your core argument for this paper is lines 188-198. The argument that sampling variance needs to be explicitly accounted for in SNP frequency estimates for pool-seq. I think this is 'known' from the literature but what I think might be under appreciated is the impact that failing to account for the sampling variance has.

"The power limitations of case-control comparisons using pool-seq are well-established (Baldwin-Brown et al. 2014; Kofler and Schlotterer 2014): to reliably detect allele frequency differences <10%, pools of thousands of individuals sequenced to >1000X average coverage are likely required"

it is not clear to me whether the papers cited are pool-seq papers where the sampling variance has been ignored or these are GWAS papers with a nominal threshold. The failure to account for multiple testing in GWAS on individual data is I think a different issue than the topic that appears to be the subject of this paper.

Paragraph starting on Line 150- poor power is a confusing, and erroneous, argument. If you are detecting SNPs these then it is not a power issue (too high type 2 error), but a false positive rate (too high type 1 error), due to sampling error. Same error line 181/182 the detection is due to elevated false positive rates - not poor power. Line 188 again false positive rate not power.

paragraph 214-226. This is worth unpacking further. What kinds of experiments use pool-seq? This paper applies only to pool-seq. Are there pool-seq experiments where haplotypes are not estimable? The papers that I am familiar with are mostly Drosophila, all from some founder population and all use a haplotype estimation. If you thus fail to account for the sampling variance in the SNPs, and you have low LD, you then may increase your multiple testing issues. This then gets very complicated and the interplay between the type 1 and type II error might conceivably come into play but this scenario is not explored in the simulation so the conclusion presented is conjecture.

paragraph 228. This is a comparison of GWAS on individual data between two types of population structures the MPP and the single population sample. I do not think this is about accuracy of allele frequency estimates due to pool-seq so it does not seem to fit well here. Further, the differences between these two designs includes the number of alleles surveyed. Consider just the null allele / genetic background effects classic example used in modifier screening (among other things). Everything depends on the allelic combinations surveyed.

Minor

The simulation seems to me to be 400 (less than recommended), 1000 (recommended coverage), 5000 (exceeds prior recommendations) this could be explained a bit better. I think the result is that the prior recommendations may be 'optimistic' and that even more coverage than previously thought may be needed.

Line 97- the word 'never' is perhaps true at this point but nonetheless a bit provocative.

Paragraph 200-212 total counts vs 2N, should be in methods and demonstrated mathematically not 'explained' heuristically.

Reviewer #3 :

I apologise that my review is late - I did not realise I would need to completely re-evaluate the paper. The revised manuscript is much clearer and better than the original. I think the MS makes a quite important point and certainly describes a better way of QTL mapping in poolseq experiments in multiparental populations than by just using naively estimated SNP dosages. To be clear, I like the paper.

However, neither in their response nor in the revision, do the authors really address many of the points in my original review and the revised manuscript still has some issues.

1. I disagree that the paper can be understood without at least knowing the basic idea behind the X-QTL methodology. The manuscript should contain a clear summary of the method. There are some details in the methods but not enough to really understand how the haplotype frequencies are being estimated. I think the method first estimates haplotype dosages in each pool at each locus by some kind of moving average (why not use an HMM?), and then estimates the SNP dosages by the expected number of reference alleles using the allelic states of the founder haplotypes. Some equations would help - eg I have no idea what a Hadamard product is.
2. In the abstract the statement "directly ascertained SNP counts have errors proportional to sequencing coverage" needs clarification. What does sequencing coverage mean exactly? It does not seem plausible to me that decreasing coverage would improve accuracy, but that is what a simple reading of that statement would suggest. Possibly they mean that the coverage should be sufficiently less than the number of chromosomes in the pool so that the probability of sampling the same chromosome twice is negligible.
3. In general the MS has too many unqualified and ambiguous statements of this nature. In particular there are a large number of assertions in the key paragraph starting on line 150. These assertions may well be true but the authors don't justify them. They need to do so. Also, I am not sure I would describe the problem of the directly ascertained SNP approach as solely due to poor statistical power *sensu stricto* (Line 150). The cloud of isolated false positive hits in Fig 3A is more likely to be due to the problem of misaligning short reads (eg due to repeats) giving rise to random fluctuations in apparent allele frequencies. This is a bioinformatics problem, not a statistical one. The haplotype based imputation approach effectively smooths out these false positives and as a bonus estimates the allele frequencies more accurately (which is a gain in power).
4. It is surely important to know whether the improved performance of the haplotype method compared to SNP association is specific to X-QTL (specifically how the haplotype frequencies are estimated) or is more generally applicable. In particular, if one does not know the founder haplotypes (eg one is dealing with an outbred population collected from the wild), can the haplotypes be estimated as part of the procedure (eg as the fast-phase and STITCH algorithms do using expectation maximisation).
5. I partially disagree with the sentence starting on line 195. One reason haplotype based approaches can out-perform SNPs is when the QTL effect is not reducible to a biallelic contrast, ie more than two allelic states are required. In the example given in the paper that is not the case ie the QTLs are biallelic so some, but not all, SNPs are good proxies.
6. I don't understand what the methodological difference is between Fig 3 and Suppl Fig 1

7. Line 200 I have tried unsuccessfully to follow the argument in this paragraph. It's obviously important. I think they are saying the apparent logP values are inflated (which might well be correct) but I am not sure I understand why. I suppose what you want to know are, at each locus, the estimated haplotype effects and their true standard errors, and then test for differences between them.
8. Minor Point. Line 76-78, the reference to the Wellcome Trust Case control consortium p-value threshold of $1e-6$ is very old. Generally a significance threshold of $5e-8$ is used in human GWAS. Not sure how relevant this is to fly poolseq experiments which presumably have a very different LD structure to human cohorts.
9. Line 82 onwards, I would suggest that another major power limitation of poolseq depends on the genetic architecture of the trait - if it is complex with many QTL then pooling the individuals with extremely high or low phenotypes won't efficiently separate out individuals with different genotypes at a given QTL, resulting in lower power.
10. Line 100 LOD should be logP.

Associate Editor Comments:

In addition to the comments above, the paper seems to argue for the more coverage than previously thought in pool-seq. This seems like a key point for readers to take home and could be made more prominent. It would also be helpful to explicitly clarify the relationship between pool-seq and GWAS on individuals for the readers.

Below is our response to the reviewer's and AE's comments. The revised manuscript has all changes in red. The comments were extremely helpful and resulted in an improved manuscript.

Reviewer #1 :

I think your argument is this: directly ascertaining SNPs and using the SNP frequency from pool-seq underestimates the sampling variance (it is the wrong test statistic). When the sampling variance is properly accounted for- either by adopting a haplotype test based on windows, or by correcting the SNP estimates then the false positive rate is controlled.

That is I think your core argument for this paper is lines 188-198. The argument that sampling variance needs to be explicitly accounted for in SNP frequency estimates for pool-seq. I think this is 'known' from the literature but what I think might be under appreciated is the impact that failing to account for the sampling variance has.

"The power limitations of case-control comparisons using pool-seq are well-established (Baldwin-Brown et al. 2014; Kofler and Schlotterer 2014): to reliably detect allele frequency differences <10%, pools of thousands of individuals sequenced to >1000X average coverage are likely required"

it is not clear to me whether the papers cited are pool-seq papers where the sampling variance has been ignored or these are GWAS papers with a nominal threshold. The failure to account for multiple testing in GWAS on individual data is I think a different issue than the topic that appears to be the subject of this paper.

Our core argument - now highlighted in sentences 2-4 of the abstract - is:

When employed in a Genome Wide Association Study (GWAS) framework, pool-seq faces a fundamental power limitation: Errors on allele frequency estimates are inversely proportional to sequence coverage and are large at modest coverage levels. Although this power limitation is widely appreciated, pool-seq GWAS lacking unambiguous hits are often interpreted as showing a highly polygenic genetic architecture. We illustrate the limitation of inferring architecture from Manhattan plots using empirical data from a *Drosophila* zinc resistance mapping study.

The core argument is that SNP-by-SNP pool-seq GWAS studies are limited in power by sequencing coverage. At the coverages commonly employed power is low, as a result of the binomial sampling errors on frequency estimates in the pools being high. While it is generally understood that power is limited in this situation (i.e., human GWAS studies employ thousands of cases versus controls!), this has not really stopped investigators from carrying out low-powered studies and interpreting Manhattan plots as demonstrating that a trait shows a polygenic architecture (we cite examples in the manuscript). **We argue that one cannot infer architecture from Manhattan plots, especially in poorly powered studies.**

The main contribution of this manuscript is providing an empirical example of this phenomenon. We show that even at 700X-plus sequencing coverage in pooled case control samples (such coverages are rarely/never achieved in the literature) a GWAS on

directly ascertained SNP frequencies failed to uncover major genes contributing to a quantitative trait (major loci that we do see with a haplotype-based approach).

We think the reviewer is perhaps focusing on the idea that haplotype inference could somehow improve case-control studies on outbred samples. But we are not advocating or proposing a solution to the problem we highlight; our contribution here is to clearly demonstrate that genetic architecture cannot be inferred from the low power, SNP-by-SNP pool-sequencing based GWAS that are common in the literature. We have now further clarified this in the manuscript (including in the abstract).

Paragraph starting on Line 150- poor power is a confusing, and erroneous, argument. If you are detecting SNPs these then it is not a power issue (too high type 2 error), but a false positive rate (too high type 1 error), due to sampling error. Same error line 181/182 the detection is due to elevated false positive rates - not poor power. Line 188 again false positive rate not power.

The central point revolves around the power issue. At a p-value threshold appropriate for the number of tests carried out, directly ascertained SNP frequencies fail to identify regions of large effect on the phenotype (Figure 3A). If one more accurately estimates the allele frequencies in the case and control pools (as we do via haplotype inference and imputation) the major regions are easily uncovered (Figure 3C). The fact that we employ a multiparental panel (MPP), and have methods for estimating haplotype frequencies is useful. But the same effect could have been achieved by just genotyping each individual in the pool to achieve more accurate allele frequency estimates (i.e., increasing N). This is a classic power problem.

The directly ascertained allele frequencies also seem to generate false positives in Figure 3A. Perhaps this is what the reviewer (the other reviewer as well) is concerned with. We alluded to this problem in our description of the hits, but this was not the main point of this manuscript, so we incorrectly moved on rather quickly. The false positives are not a feature of the simulations (Figure 1), leading us to believe the false positives are likely strange genotyping errors which may or may not be properly controlled for in the literature. Genotyping errors are likely a real problem in Illumina libraries sequenced to extremely high coverage. We have now tried to clarify this in the manuscript when we describe the initial GWAS on directly ascertained SNPs.

paragraph 214-226. This is worth unpacking further. What kinds of experiments use pool-seq? This paper applies only to pool-seq <not to GWAS on individual data>. Are there pool-seq experiments where haplotypes are not estimable? The papers that I am familiar with are mostly Drosophila, all from some founder population and all use a haplotype estimation. If you thus fail to account for the sampling variance in the SNPs, and you have low LD, you then may increase your multiple testing issues.

Pool-seq experiments are often applied to outbred samples where haplotype estimation is impossible (or not employed). This is especially prevalent in Evolve and Resequencing

experiments. But also case-control type genetic mapping experiments. We cite several examples claiming polygenic architecture from Manhattan plots (line 87 of the original paper).

Despite this constraint, Manhattan plots of $-\log_{10}(\text{p-value})$ against genomic location from experiments with much lower sequence coverage, and that show little evidence of "hits", are often interpreted as representing a highly polygenic architecture (Huang et al. 2012; Orozco-terWengel et al. 2012; Morozova et al. 2015; Fochler et al. 2017; Zhou et al. 2017; Barghi et al. 2019; Lirakis et al. 2022).

This then gets very complicated and the interplay between the type 1 and type II error might conceivably come into play but this scenario is not explored in the simulation so the conclusion presented is conjecture.

This outbred population situation is exactly the case examined in the simulation where we use MSPrime to simulate a 1Mb neutrally evolving segment of the genome under *D. melanogaster* natural population type parameters. The simulation shows the power of case control studies at an appropriate p-value threshold to control for false positives, and illustrates that the power to detect true associations is low unless pools are sequenced to extremely high coverage. The simulation largely agrees with the consensus in human genetics that three thousand fully genotyped samples are required to detect sites contributing ~5% to complex trait variation.

This is exactly the point of the paper. Experiments are being carried out on outbred type samples where haplotype estimation is not possible. With SNP-by-SNP tests being carried out at modest levels of coverage. These are low powered studies to detect even large allele frequency shifts. Yet the resulting Manhattan plots are being interpreted as supporting a highly polygenic architecture.

paragraph 228. This is a comparison of GWAS on individual data between two types of population structures the MPP and the single population sample. I do not think this is about accuracy of allele frequency estimates due to pool-seq so it does not seem to fit well here. Further, the differences between these two designs includes the number of alleles surveyed. Consider just the null allele / genetic background effects classic example used in modifier screening (among other things). Everything depends on the allelic combinations surveyed.

We agree with the sentiment of the reviewer. But this is the last paragraph of the paper, so we were hopeful we would be allowed some speculation. It is easy to extrapolate from this paper that the community is perhaps overthinking things a little. Although all these other "forces" are at play (number of alleles, background, GEI, epistasis, ... you name it). This may all be minor relative to poorly powered experiments. Genotyping a few hundred RILs (for instance) may be such a compromised experimental design in terms of power, that all these other factors just don't matter. We think the community should ask itself if the traits truly are polygenic with individual alleles explaining <<1% of variation, why are we genotyping (a massively insufficient) 200 RILs? If alleles

contribute 5% to complex trait variation, why are we genotyping a similarly insufficient 200 RILs? In neither case are we in a strong position to make useful inferences about architecture. We think the speculation is OK at this point in the manuscript.

Minor

The simulation seems to me to be 400 (less than recommended) , 1000 (recommended coverage) , 5000 (exceeds prior recommendations) this could be explained a bit better. I think the result is that the prior recommendations may be 'optimistic' and that even more coverage than previously thought may be needed.

We have clarified the choices of coverage in the text. At the lower end these are coverages that could be routinely achieved in these types of experiments. The upper end approximates 2500 fully genotyped cases or controls (in terms of error/power).

Line 97- the word 'never' is perhaps true at this point but nonetheless a bit provocative.

We are happy to delete, and actually debated the use of the word. The important point is that the empirical coverages we explore (>700X) are just not achieved in the literature. So the actual situation is likely worse than what we illustrate.

Paragraph 200-212 total counts vs 2N, should be in methods and demonstrated mathematically not 'explained' heuristically.

We have modified the methods and discuss this further below in response to reviewer #3.

Reviewer #3 :

I apologise that my review is late - I did not realise I would need to completely re-evaluate the paper. The revised manuscript is much clearer and better than the original. I think the MS makes a quite important point and certainly describes a better way of QTL mapping in poolseq experiments in multiparental populations than by just using naively estimated SNP dosages. To be clear, I like the paper.

We thank the reviewer for their positive comments. Although we think there is some confusion that we hope our revisions clarify. We are not proposing that haplotype estimates can be used to derive more accurate estimates of SNP frequencies in pool-seq experiments in a general manner. We do not claim this, and point out the opposite in the discussion. The dataset of our example is somewhat unique, and allows us to do this (and such an approach does have utility in pool-seq experiments employing MPPs). By exploiting this pool-seq dataset obtained from a MPP we are able to show that even at >700X sequence coverage directly ascertained SNP-based GWAS are poorly powered to detect major effect genes.

However, neither in their response nor in the revision, do the authors really address many of the points in my original review and the revised manuscript still has some issues.

1. I disagree that the paper can be understood without at least knowing the basic idea behind the X-QTL methodology. The manuscript should contain a clear summary of the method. There are some details in the methods but not enough to really understand how the haplotype frequencies are being estimated. I think the method first estimates haplotype dosages in each pool at each locus by some kind of moving average (why not use an HMM?), and then estimates the SNP dosages by the expected number of reference alleles using the allelic states of the founder haplotypes. Some equations would help - eg I have no idea what a Hadamard product is.

We added a paragraph to the methods.

The MPP we study is ultimately derived by combining 8 known founders. We estimate, for windows of a few hundred kb, the mixture of founder haplotypes most consistent with the observed vector of SNP frequencies in the pooled sample. This approach is well-described in the literature and appropriate papers are cited. To additionally estimate SNP frequencies we sum over the element-wise product of the vector of haplotype frequencies and the vector of founder SNP states. An HMM is not so appropriate for this task as one first needs to estimate the underlying vector of eight frequencies (which feels more like a least squares problem). An HMM would be better suited to estimating a genotype which can take on 3 possible states (AA, Aa, or aa) than a vector of frequencies. The constrained least squares approach we use works quite well and is widely used. Furthermore it produces a variance covariance error matrix associated with estimates (which can be exploited moving forward).

Hadamard product = "element-wise product" of two vectors. Although we do not use this term anymore.

2. In the abstract the statement "directly ascertained SNP counts have errors proportional to sequencing coverage" needs clarification. What does sequencing coverage mean exactly? It does not seem plausible to me that decreasing coverage would improve accuracy, but that is what a simple reading of that statement would suggest. Possibly they mean that the coverage should be sufficiently less than the number of chromosomes in the pool so that the probability of sampling the same chromosome twice is negligible.

The reviewer is correct, the abstract should have read "inversely proportional to ... and large at modest levels of coverage". We have now corrected this.

If coverage is much less than the number of individuals in the pool, then the error on SNP frequencies are binomial in coverage. We cite the Futschik paper in the first paragraph of the paper claiming this. In theory then, increasing coverage will result in more precise estimates of allele frequency as long as coverage is much less than twice the number of individuals in the pool. In practice allele frequency estimates are slightly overdispersed, and one could imagine the quality of the Illumina library begins to matter at coverages exceeding 100-200X. We have modified the abstract.

3. In general the MS has too many unqualified and ambiguous statements of this nature. In particular there are a large number of assertions in the key paragraph starting on line 150. These assertions may well be true but the authors don't justify them. They need to do so. Also, I am not sure I would describe the problem of the directly

ascertained SNP approach as solely due to poor statistical power sensu stricto (Line 150). The cloud of isolated false positive hits in Fig 3A is more likely to be due to the problem of misaligning short reads (eg due to repeats) giving rise to random fluctuations in apparent allele frequencies. This is a bioinformatics problem, not a statistical one. The haplotype based imputation approach effectively smooths out these false positives and as a bonus estimates the allele frequencies more accurately (which is a gain in power).

We have attempted to clarify this in the text as both reviewers had problems here. When we do the scan on directly ascertained SNPs we observe two things. First, we fail to observe obvious hits in the regions of the genome where major genes affecting zinc-resistance were mapped by Hansen et al. (PMID: 40839771) This is the observation we focus on as it suggests poor power even at >700X sequencing coverage. And second we get these sporadic hits that are likely false positives, that we did not properly discuss (and both reviewers note).

We have now rewritten this paragraph. We agree with the reviewer that these false positives are likely going to be common if you sequence Illumina libraries to very high coverage and likely represent regions where reads are misaligned and SNPs are being improperly called. These artifacts are something the field should work on, and may indeed be better controlled in studies other than this one. But, these potential false positive hits are tangential to the arguments being made in this short note. The key observation is that we did not observe clear signals indicating "hits" at 700X coverage.

4. It is surely important to know whether the improved performance of the haplotype method compared to SNP association is specific to X-QTL (specifically how the haplotype frequencies are estimated) or is more generally applicable.

We do not believe the haplotype method to be generally applicable. It is only going to work if you have a MPP you are studying. There was a sentence to this effect in the discussion that we have now expanded. We feel that both reviewers want this paper to be about proposing an improved SNP imputation method that is generally applicable to outbred samples. But we make clear throughout the manuscript the imputation method is only appropriate to MPPs. The paper claims that this dataset is a unique case where we can impute haplotypes and SNP frequencies with considerable accuracy. We have now stressed this in the abstract as well.

What we show is that given accurate estimates of SNP or haplotype frequencies we uncover a relatively simple genetic architecture. Whereas directly ascertained SNP frequencies (even at coverages much higher than are really ever employed in the literature) fail to uncover this simple architecture. We are led to conclude that many studies are poorly powered and can only conclude a polygenic architecture, even when a more simple architecture exists. As an example this demonstration is rather unique, as it does not require comparing across traits or studies, nor is it a simulation. We think

it clearly demonstrates an example where directly ascertained SNP-by-SNP GWAS fails to uncover a relatively simple architecture.

In particular, if one does not know the founder haplotypes (eg one is dealing with a outbred population collected from the wild), can the haplotypes be estimated as part of the procedure (eg as the fast-phase and STITCH algorithms do using expectation maximisation).

Perhaps STITCH could be modified (we have used it extensively in our mouse work and love it). But STITCH estimates diploid genotypes when applied to single individuals. It is not designed for estimating SNP frequencies in a pooled sample. So we do not speculate among these lines.

5. I partially disagree with the sentence starting on line 195. One reason haplotype based approaches can out-perform SNPs is when the QTL effect is not reducible to a biallelic contrast, ie more than two allelic states are required. In the example given in the paper that is not the case ie the QTLs are biallelic so some, but not all, SNPs are good proxies.

We agree 100%. There are clearly cases where haplotype-based tests can outperform SNP-based tests. But in this example the real advantage of the haplotype-based tests is simply more power coming from more accurate frequency estimates. This is illustrated in Figure 3C, where SNP-frequency estimates derived from the haplotype estimates perform comparably to the haplotype-based tests. And both massively outperform directly ascertained SNPs. So the haplotype- versus SNP-based tests is perhaps a second order effect, and the real gain just comes from accurate estimates of frequency, period. We have now clarified this in the paper.

6. *I don't understand what the methodological difference is between Fig 3 and Suppl Fig 1*

None at all. We are merely zooming in on a region. We were attempting to anticipate that a reader may think the Manhattan plots - by virtue of showing the entire genome - obscure important details located at QTL. We attempted to clarify in text and figure legend.

7. *Line 200 I have tried unsuccessfully to follow the argument in this paragraph. It's obviously important. I think they are saying the apparent logP values are inflated (which might well be correct) but I am not sure I understand why. I suppose what you want to know are, at each locus, the estimated haplotype effects and their true standard errors, and then test for differences between them.*

We rewrote what was clearly a poor description in the methods (Reviewer 1 had the same issue). To make the 3 scans (directly ascertained, imputed haplotypes, and imputed SNPs) comparable we carry out CMH tests in all 3 cases. CMH-tests are essentially replicated chi-square tests of REF vs ALT counts (or HAP1, HAP2, ..., HAP8) against case vs control status. In the case of the directly ascertained SNPs we directly obtain counts. In the other two cases we have frequency estimates that we have to convert to counts, which we do by multiplying the frequencies by twice the number of

individuals (i.e., the number of alleles) in the pool. This would be correct if each individual were directly genotyped as would be typical in a human GWAS. In reality the efficiency of our haplotype estimate procedure is not 100% and we really should be multiplying frequencies by $k \cdot 2N$ where k is the efficiency of the imputation approach. Chi-square test statistics on frequencies as well as $-\log_{10}(p\text{-values})$ scale by a multiplicative constant in sample size. The resulting net impact is that the y-axis for figure 3B and 3C is inflated by an unknown multiplicative constant. The inflation factor is unknown but we guess $\sim 30\%$ from experiments where we have estimates of allele frequency from very high coverage samples we can compare to imputed frequencies.

8. Minor Point. Line 76-78, the reference to the Wellcome Trust Case control consortium p-value threshold of $1e-6$ is very old. Generally a significance threshold of $5e-8$ is used in human GWAS. Not sure how relevant this is to fly poolseq experiments which presumably have a very different LD structure to human cohorts.

We discussed referring to the more modern threshold of $1e-8$ used in human GWAS before submission, but prefer to stick with the original WTCCC $1e-6$. We are trying to avoid a secondary issue in the fly DGRP literature where a threshold of $1e-5$ is used as a "suggestive threshold", despite $\sim 3M$ tests being carried out in flies, smaller panels being used than humans, and flies exhibiting much lower levels of LD than humans. We have also avoided a threshold in Figure 1 for exactly the same reason. The point with respect to this paper is that some stringent threshold is required, and even at a "liberal threshold", there are major challenges in detecting true allele frequency differences of a few percent.

We are inclined to leave this as is. But if the AE wishes us to add a sentence referring to the $1e-8$ we are equally OK.

9. Line 82 onwards, I would suggest that another major power limitation of poolseq depends on the genetic architecture of the trait - if it is complex with many QTL then pooling the individuals with extremely high or low phenotypes won't efficiently separate out individuals with different genotypes at a given QTL, resulting in lower power.

We are unwilling to make this claim without more analytical work or simulations. Height is highly polygenic, yet tall people still have tall kids, right? So extreme mapping will tend to enrich for high-alleles in the tails of the population. **Of course, the allele frequency differences between the extreme pool and the control pool may be too subtle to detect.** The reviewer's point only further supports the claim of this work. It is important to consider how large your panel is and how well-sequenced it is in terms of what you are trying to detect. A human GWAS with 3K fully genotyped cases and controls employing a 1M feature SNP chip that detects nothing is strong evidence that the trait is polygenic, in the sense that there are unlikely to be *many* SNPs contributing 5% to trait variation that are going undetected. A more modestly powered experiment

cannot show this and Manhattan plots from such studies cannot be interpreted as supporting a polygenic architecture.

10. Line 100 LOD should be $\log P$.

Fixed.

Associate Editor Comments:

In addition to the comments above, the paper seems to argue for the more coverage than previously thought in pool-seq. This seems like a key point for readers to take home and could be made more prominent. It would also be helpful to explicitly clarify the relationship between pool-seq and GWAS on individuals for the readers.

We have modified the text and the abstract. In a case-control with samples obtained from a large outbred population power comes from the counts in the SNP-by-SNP chi-square tables. As a result, a pool-seq case-control sequenced to $2N$ coverage, will have power roughly comparable to a fully genotyped case-control consisting of N individuals. Directly ascertained SNP-by-SNP pool-seq case-control experiments characterized at low coverage levels lack sufficient power to draw inferences about trait architecture.

January 26, 2026

RE: GENETICS-2025-308929

Dear Dr. Long:

I am pleased to accept your manuscript titled "The Illusion of Polygenicity in Poolseq studies: Insufficient Power Can Mask Simple Genetic Architectures" for publication in GENETICS, pending minor revision.

Thank you for the revisions of this work, which both reviewers and I agree has strengthened it. Each reviewer had one remaining suggestion that I hope you can quickly and easily address: (1) add a 'cartoon' figure explaining the expected patterns of type 1, type 2 errors in a pool -seq GWAS. (2) remove the reference to the 1e6 threshold in the WTCCC. I expect you should be able to submit a revised manuscript within 30 days. A suitably revised manuscript will be acceptable for publication; I don't expect to send it out for review.

When revising the ms., please make an effort to shorten it, because that almost always improves a manuscript. We urge authors to heed the advice of Strunk and White: "omit needless words"¹. Follow this link to submit the revised manuscript: Link Not Available

Thank you for submitting this story to Genetics.

Sincerely,

Patricia Wittkopp
Associate Editor
GENETICS

Approved by:
Mario Calus
Senior Editor
GENETICS

Reviewer comments:

Reviewer #1 :

Thank you for the clarifications.

Despite the clarifying text, I do wonder if a 'new' reader will appreciate the 'red herring' that is the false positives that distracted both myself and R3.

I think the reader would benefit from a 'cartoon' figure that explained the expected patterns of type 1, type 2 errors in a pool -seq GWAS.

Reviewer #3 :

I think the authors have satisfactorily addressed my questions, and the paper is much clearer regarding the authors' purpose and conclusions.

Although one could go on nit-picking, this review process is becoming more like a discussion among friends than a constructive process so I won't make any other suggestions except that I think they should remove the reference to the 1e6 threshold in the WTCCC as its not needed and would put off human geneticists.

Response to Reviewer Concerns

We have addressed both reviewer concerns raised in the previous round of review:

Reviewer #1 requested a 'cartoon' figure explaining the expected patterns of type 1 and type 2 errors in a pool-seq GWAS to help readers appreciate the 'red herring' of false positives.

We have added Supplementary Figure 1, which illustrates how true signals can be distinguished from alignment artifacts as a function of sequencing coverage and linkage disequilibrium scale. This figure is referenced in the main text (lines 124-137), and the previous Supplementary Figure 1 has been renumbered to Supplementary Figure 2.

Reviewer #3 suggested removing the reference to the 1×10^{-6} threshold in the WTCCC study.

We have revised lines 78-81 to present a range of commonly used significance thresholds (10^{-6} to 10^{-9}) without specific citation, avoiding potential concerns from human geneticists while retaining relevant context.

February 19, 2026

RE: GENETICS-2025-308929R1

Dr. Anthony D. Long
University of California, Irvine
Department of Ecology and Evolution
Steinhaus Hall
Irvine, California 92697-2525

Dear Dr. Long:

Congratulations, your manuscript titled "The Illusion of Polygenicity in Poolseq studies: Insufficient Power Can Mask Simple Genetic Architectures" is accepted for publication in GENETICS! Many thanks for submitting your research to the journal.

To Proceed to Publication:

1. Format your article according to GENETICS style: <https://academic.oup.com/genetics/pages/author-guidelines>
2. Ensure that you comply with data and community resource citation guidelines: <https://academic.oup.com/genetics/pages/author-guidelines#section-5-9-2>
3. Upload your final files at <https://genetics.msubmit.net>
4. Add oupsupport@scipris.com and genetics.oup@novatechset.com (or the domains @scipris.com and @novatechset.com) to your email program's "safe senders" list. You will be contacted by both at various points during the production process.

Notes:

- Your currently-accepted manuscript (unedited, as submitted, reviewed, and accepted) will be published at GENETICS and deposited into PubMed as an Advance Access article. Notify sourcefiles@thegsajournals.org before signing your license if you do not wish to publish your article via Advance Access.
- We invite you to submit an original color figure related to your paper for consideration as cover art. Please email your submission to the editorial office or upload it with your final files. You can submit a small-sized image for evaluation, and if selected, the final image must be a TIFF file 2513px wide by 3263px high (8.375 by 10.875 inches; resolution of 600ppi). Please avoid graphs and small type.
- After files are sent to Oxford University Press we use SciPris to manage article licensing and payment. If you do not have a SciPris account, you will receive an email from no-reply@scipris.com to sign up to use Oxford University Press' author portal. After logging in, follow the online instructions to sign your license and arrange any payment due.

If you have any questions or encounter any problems while uploading your accepted manuscript files, please email the editorial office at sourcefiles@thegsajournals.org.

Sincerely,

Patricia Wittkopp
Associate Editor
GENETICS

Approved by:
Mario Calus
Senior Editor
GENETICS